# Establishment of a Quadruplex RT-qPCR for the Detection of Canine Coronavirus, Canine Respiratory Coronavirus, Canine Adenovirus Type 2, and Canine Norovirus

**DOI:** 10.3390/pathogens13121054

**Published:** 2024-11-29

**Authors:** Kaichuang Shi, Yandi Shi, Yuwen Shi, Feng Long, Yanwen Yin, Yi Pan, Zongqiang Li, Shuping Feng

**Affiliations:** 1School of Basic Medical Sciences, Youjiang Medical University for Nationalities, Baise 533000, China; panyiyi2004@163.com; 2College of Animal Science and Technology, Guangxi University, Nanning 530005, China; shiyandi123@126.com (Y.S.); shiyuwen2@126.com (Y.S.); zqingli@gxu.edu.cn (Z.L.); 3Guangxi Center for Animal Disease Control and Prevention, Nanning 530001, China; longfeng1136@163.com (F.L.); yanwen0349@126.com (Y.Y.)

**Keywords:** canine coronavirus (CCoV), canine respiratory coronavirus (CRCoV), canine adenovirus type 2 (CAV-2), canine norovirus (CNV), quadruplex RT-qPCR

## Abstract

Canine coronavirus (CCoV), canine respiratory coronavirus (CRCoV), canine adenovirus type 2 (CAV-2), and canine norovirus (CNV) are important pathogens for canine viral gastrointestinal and respiratory diseases. Especially, co-infections with these viruses exacerbate the damages of diseases. In this study, four pairs of primers and probes were designed to specifically amplify the conserved regions of the CCoV M gene, CRCoV N gene, CAV-2 hexon gene, and CNV RdRp gene. After optimizing different reaction conditions, a quadruplex RT-qPCR was established for the detection of CCoV, CRCoV, CAV-2, and CNV. The specificity, sensitivity, and repeatability of the established assay were evaluated. Then, the assay was used to test 1688 clinical samples from pet hospitals in Guangxi province of China during 2022–2024 to validate its clinical applicability. In addition, these samples were also assessed using the reported reference RT-qPCR assays, and the agreements between the developed and reference assays were determined. The results indicated that the quadruplex RT-qPCR could specifically test only CCoV, CRCoV, CAV-2, and CNV, without cross-reaction with other canine viruses. The assay had high sensitivity with limits of detection (LODs) of 1.0 × 10^2^ copies/reaction for CCoV, CRCoV, CAV-2, and CNV. The repeatability was excellent, with intra-assay variability of 0.19–1.31% and inter-assay variability of 0.10–0.88%. The positivity rates of CCoV, CRCoV, CAV-2, and CNV using the developed assay were 8.59% (145/1688), 8.65% (146/1688), 2.84% (48/1688), and 1.30% (22/1688), respectively, while the positivity rates using the reference assays were 8.47% (143/1688), 8.53% (144/1688), 2.78% (47/1688), and 1.24% (21/1688), respectively, with agreements of more than 99.53% between two methods. In conclusion, a quadruplex RT-qPCR with high sensitivity, specificity, and repeatability was developed for rapid, and accurate detection of CCoV, CRCoV, CAV-2, and CNV.

## 1. Introduction

The number of pet dogs in different countries around the world has been constantly increasing. However, various diseases threat the health of dogs, of which viral respiratory and gastrointestinal infections of canine coronavirus (CCoV), canine respiratory coronavirus (CRCoV), canine adenovirus type 2 (CAV-2), and canine norovirus (CNV) are common in dogs [1,2,3,4]. Especially, co-infections of two and more than two of these viruses with other gastrointestinal/respiratory related viruses are usually found in clinical samples, which can exacerbate the diseases with serious clinical signs and pathological damages [3,5,6,7]. Therefore, accurate and reliable detection and differentiation of these pathogens is vital for diagnosis and treatment of these diseases.

CCoV is an enveloped, single-stranded, positive-sense RNA virus with a genome size of 28–32 kb, and is classified in the genus *Alphacoronavirus* of the family *Coronaviridae* [8]. CCoV was first reported in 1971, and can affect gastrointestinal/respiratory tract with mild localized disease, but may also lead to systemic infections [9]. CCoV is divided into two genotypes, i.e., genotype I and II, and genotype II is further divided into genotype IIa and IIb. CRCoV is classified in the genus *Betacoronavirus* of the *Coronaviridae* family. It was first found in 2003 in the United Kingdom [10] and is thought to have originated from bovine coronavirus (BCoV) [11]. The infected dogs usually experience cough, sneeze, and runny noses, while severe infections progress to pneumonia [2]. CNV is a small icosahedral, non-enveloped, positive-sense RNA virus with a genome size of approximately 7.7 kb, which is classified in the genus *Norovirus* of the family *Caliciviridae* [12]. CNV was first discovered in 2007 in an Italian puppy with gastroenteritis [13]. To date, genotypes GIV, GVI, and GVII of CNV have ever been reported. CAV belongs to the genus *Mastadenovirus* of the family *Adenoviridae*, and is a linear, non-segmented, double-stranded DNA virus with a full-genome length of 20–30 kb [14]. CAV is classified into type 1 (CAV-1) and type 2 (CAV-2), of which CAV-1 induces infectious canine hepatitis (ICH), and CAV-2 causes infectious tracheobronchitis (ITB) and enteritis [14]. CAV-2 was first detected in dogs with signs of upper respiratory tract infection at a Canadian shelter in 1961 [15]. Usually, the clinical symptoms of CAV-2 infection alone are rarely apparent or mild, while co-infection with other viruses exacerbates the symptoms of CAV-2 infection with typical ITB clinical signs, resulting in severe paroxysms of coughing and runny nose [2,16]. The cough is usually accompanied by mucous secretions, and at its worst the condition is progressed to bronchopneumonia, which can lead to death.

CRCoV and CAV-2 are both common viruses that cause canine infectious respiratory disease complex (CIRDC), and are widely distributed worldwide [2,3,6,7,17,18,19,20]. Dogs infected with CCoV and CNV can develop acute intestinal infections with clinical signs of vomiting, diarrhea, dehydration, and CCoV and CNV have been found in many countries around the world [21,22,23,24]. Especially, CCoV causes gastrointestinal/respiratory signs [9], and CAV-2 causes infectious tracheobronchitis (ITB) and enteritis [14], which leads to confusion between the two diseases in clinical practice. In addition, it has shown that two or more of CCoV, CRCoV, CAV-2, and CNV can be detected simultaneously in dogs with different clinical signs [5,6,7], and co-infections of these pathogens exacerbate the seriousness of the diseases. The rapid, and accurate detection and diagnosis of these viral infections is vital to provide prompt and effective treatment. The multiplex real-time quantitative RT-PCR (RT-qPCR) shows advantages of high sensitivity, specificity, and repeatability, and can detect several kinds of nucleic acids at the same time in one reaction [25]. This technique has been widely used to test viral pathogens in different laboratories [26]. The multiplex RT-qPCR methods have been established to test one or two of CCoV, CRCoV, CAV-2, and CNV, as well as other viruses [27,28,29,30]. However, to date, there is no report on RT-qPCR method that can simultaneously detect CCoV, CRCoV, CAV-2, and CNV. Therefore, a quadruplex RT-qPCR for the simultaneous detection of CCoV, CRCoV, CAV-2, and CNV was developed in this study, which showed highly sensitive, specific, rapid, and accurate.

## 2. Materials and Methods

### 2.1. Reference Strains

The following vaccine strains were purchased from Zoetis Inc. (ZTS) (Lincoln, NE, USA): CCoV (NL-18 strain), CAV-2 (Manhattan strain), canine parainfluenza virus (CPIV, NL-CPI-5 strain), canine distemper virus (CDV, Snyder Hill strain, GenBank accession No. JN896987), and canine parvovirus (CPV, NL-35-D strain, GenBank accession No. MW650832). The vaccine strains of CDV (Onderstepoort strain, GenBank accession No. AF305419), CPV (154 strain, GenBank accession No. ON479058), CPIV (Cornell strain) were purchased from Merck & Co., Inc. (Kenilworth, NJ, USA). The vaccine strains of CDV (CDV/R-20/8 strain), CPIV (CPIV/A-20/8 strain), CPV (CR86106 strain) were purchased from Keqian Biology Co., Ltd. (Wuhan, China).

### 2.2. Positive Samples

The positive clinical samples of CCoV, CPV, CDV, canine rotavirus (CRV), canine circovirus (CanineCV), CNV, CRCoV were detected using qPCR/RT-qPCR in clinical samples from pet hospitals and were confirmed by gene sequencing.

### 2.3. Clinical Samples

From January 2022 to March 2024, a total of 1688 clinical samples (including nasal swabs, anal swabs, and feces from each dog) were obtained from sick dogs with different gastroenteritis signs and/or respiratory signs (nasal swabs, anal swabs, and fecal samples from each dog were mixed together and considered as one sample for detection of viral nucleic acids). The samples were collected from 22 pet hospitals in seven cities, i.e., Nanning, Baise, Liuzhou, Guilin, Yulin, Qinzhou, and Beihai, in Guangxi province of China. All samples were delivered (≤4 °C) to our laboratory within 4 h post collection for the detection of CCoV, CRCoV, CAV-2, and CNV.

### 2.4. Extraction of Nucleic Acids

The nasal swabs, anal swabs, and about 0.5 g fecal samples collected from sick dogs were placed in 2.0 mL EP tubes with 1.5 mL of phosphate-buffered saline (PBS, pH 7.2), spun for 30 s using a vortex shaker, frozen and thawed 3 times, and centrifuged for 2 min (4 °C, 12,000 rpm). Then, 200 μL of the supernatant of clinical samples was used for nucleic acid extraction using the Nucleic Acid Extraction and Purification Kit (Zijian, Shenzhen, China). The extracted nucleic acids were used to test CCoV, CRCoV, CAV-2, and CNV immediately, or stored at −80 °C until use.

The positive clinical samples of CCoV, CPV, CDV, CRV, CanineCV, CNV, and CRCoV were used to extract nucleic acids as described above. The vaccine solutions were used to extract nucleic acids directly using the Nucleic Acid Extraction and Purification Kit (Zijian, Shenzhen, China). The obtained nucleic acids from positive clinical samples and vaccine solutions were used as positive control to analyze the specificity of the developed quadruplex RT-qPCR.

### 2.5. Primers and Probes

The genome sequences of 36 representative strains of CCoV (7 CCoV-I strains, 21 CCoV-IIa strains, and 8 CCoV-IIb strains), 31 strains of CRCoV, 44 strains of CAV-2, and 33 strains of CNV (28 strains of CNV-GVI, and 5 strains of CNV-GIV) were downloaded from the GenBank in National Center for Biotechnology Information (NCBI, https://www.ncbi.nlm.nih.gov/nucleotide/, accessed on 15 October 2021). These representative strains of CCoV, CRCoV, CAV-2, and CNV were selected based on their origination from different countries and isolation of different dates. The multiple sequence alignments were performed using MEGA-X 10.2.6 software (https://www.megasoftware.net/dload_win_gui/, accessed on 15 October 2021). Four pairs of primers and probes (Table 1) were designed targeting the conserved regions of the M gene of CCoV, the N gene of CRCoV, the hexon gene of CAV-2, and the RdRp gene of CNV, respectively, using Oligo 7.0 software (https://www.oligo.net/downloads.html, accessed on 15 October 2021). These primers and probes were used to detect different genotypes of CCoV, CRCoV, CAV-2, and CNV simultaneously. The information on the reference strains is shown in the Appendix A, and the positions of the designed primers and probes are shown in the Appendix A.

### 2.6. Construction of Standard Plasmids

The nucleic acids extracted from the positive clinical samples of CCoV, CRCoV, and CNV were reverse-transcribed into cDNA using TaKaRa PrimeScript II 1st Strand cDNA Synthesis Kit (Dalian, China). The cDNAs of CCoV, CRCoV, and CNV, and the total DNA of CAV-2 (Manhattan strain) extracted from vaccine solution were used as templates to amplify the viral target fragments with specific primers (Table 1) using TaKaRa *Taq* PCR MasterMix Kit (Dalian, China), and the amplification procedures were conducted at 94 °C for 3 min, 35 cycles of 94 °C for 30 s, 57 °C for 30 s, 72 °C for 12 s, and 72 °C for 10 min.

The products from PCR amplification were subjected to gel electrophoresis, and observed using a UV transilluminator (UVItec, Cambridge, UK). The agarose gels with the correct amplified fragments were collected and placed into 1.5 mL EP tubes, purified using TaKaRa MiniBEST Agarose Gel DNA Extraction Kit (Dalian, China), ligated into TaKaRa pMD-18T vector (Dalian, China), and transformed into *E. coli DH5α*-competent cells. The positive clones were cultured for 22–24 h at 37 °C. The recombinant plasmid constructs were extracted using TaKaRa MiniBEST Plasmid Purification Kit (Dalian, China), and named p-CCoV, p-CRCoV, p-CAV-2, and p-CNV. Their concentrations were calculated according to OD_260 nm_ and OD_280 nm_ values using the following equation: plasmid (copies/µL) = $\frac{\left(6.02\times 10^{23})\times (Xng/µL\times 10^{-9}\right)}{plasmid\;length\;(bp)\times 660}$.

### 2.7. Optimization of Reaction Conditions

The quadruplex RT-qPCR was performed using an ABI QuantStudio 5 qPCR system (Carlsbad, CA, USA). The plasmid constructs p-CCoV, p-CRCoV, p-CAV-2, and p-CNV were mixed at an equal volume (1:1:1:1) as templates, and the annealing temperature (54–60 °C), primer, and probe concentration (20 pmol/µL, 0.1–0.5 µL) were optimized. The reaction system was performed with One Step PrimeScript™ RT-PCR Kit (TaKaRa, Dalian, China) at a total volume of 20 µL to find the optimal reaction conditions. The amplification parameters were as follows: 42 °C 5 min; 40 cycles of 95 °C 10 s, 95 °C 5 s, 57 °C 30 s; the fluorescence signal was automatically recorded at the end of each cycle.

### 2.8. Formation of Standard Curves

The p-CCoV, p-CRCoV, p-CAV-2, and p-CNV was mixed at an equal ratio (1:1:1:1), and then diluted 10-fold serially from 1 × 10^10^ to 1 × 10^−1^ copies/µL. The standard curves were generated using the mixed plasmid constructs with concentration from 1 × 10^9^ to 1 × 10^2^ copies/µL (the final concentration: 1 × 10^8^ to 1 × 10^1^ copies/µL) as templates, and performed qPCR using the optimized reaction conditions.

### 2.9. Evaluation of Specificity

The RNA/DNA of CCoV (NL-18 strain), CAV-2 (Manhattan), CDV (Snyder Hill, Onderstepoort, and CDV/R-20/8 strains), CPIV (NL-CPI-5, Cornell, and CPIV/A-20/8 strains), CPV (NL-35-D, 154, and CR86106 strains), CRV, and CanineCV were used as templates to assess the specificity of the quadruplex RT-qPCR. The mixture of p-CCoV, p-CRCoV, p-CAV-2, and p-CNV was used as positive control, and the negative clinical sample and nuclease-free distilled water were used as negative controls.

### 2.10. Assessment of Sensitivity

The p-CCoV, p-CRCoV, p-CAV-2, and p-CNV were mixed at an equal ratio (1:1:1:1), diluted 2-fold serially from 400, 200, 100, to 50 copies/reaction, and used as templates to evaluate the LODs of the quadruplex RT-qPCR assay. Probit regression analysis was used to evaluate the LOD values.

### 2.11. Assessment of Repeatability

The mixture of p-CCoV, p-CRCoV, p-CAV-2, and p-CNV was diluted 10-fold serially, and the concentrations of 1 × 10^9^, 1 × 10^6^, and 1 × 10^3^ copies/µL were used as templates to assess the assay’s repeatability. Intra-assay tests were repeated three times each and inter-assay tests were performed on three different days to calculate the coefficient of variations (CVs).

### 2.12. Test of Clinical Samples

The 1688 clinical samples from Guangxi province were tested for CCoV, CRCoV, CAV-2, and CNV using the established quadruple RT-qPCR and the reference RT-qPCR reported for the detection of CCoV [28], CRCoV [31], CAV-2 [32], and CNV [33]. The clinical sensitivity and specificity of the established assay were calculated, and the coincidence rates of the results of these assays for the detection of CCoV, CRCoV, CAV-2, and CNV were compared.

## 3. Results

### 3.1. Construction of Standard Plasmid Constructs

The cDNA/DNA from positive clinical samples/vaccine solution of CCoV, CRCoV, CAV-2, and CNV were used as templates to amplify the target fragments of CCoV M gene, CRCoV N gene, CAV-2 hexon gene, and CNV RdRp gene using the specific primers in Table 1. The PCR products were purified, ligated to PMD-18T vector, transformed into DH5α competent cells, and inoculated in LB nutrient agar medium containing ampicillin. The positive clones were selected and cultured, and the recombinant plasmid constructs were extracted. The obtained plasmid constructs were confirmed by sequence analysis, and named p-CCoV, p-CRCoV, p-CAV-2, and p-CNV. They were used as the standard plasmid constructs in the process of development of the quadruplex RT-qPCR. The initial concentrations of p-CCoV, p-CRCoV, p-CAV-2, and p-CNV were 9.28 × 10^10^, 5.35 × 10^10^, 7.0 × 10^10^, and 5.61 × 10^10^ copies/µL, respectively. They were diluted to 1.0 × 10^10^ copies/µL, and stored at −80 °C until use.

### 3.2. Determination of Reaction Conditions

The optimal reaction conditions for the quadruplex RT-qPCR were determined after several trials mapping the annealing temperature, primer and probe concentration. The 20 µL reaction system contained the ingredients as shown in Table 2. The amplification parameters were as follows: 42 °C 5 min; 40 cycles of 95 °C 10 s, 95 °C 5 s, and 57 °C 30 s. The samples with Ct values ≤ 36 were considered as positive samples.

### 3.3. Generation of Standard Curves

The p-CCoV, p-CRCoV, p-CAV-2, and p-CNV were mixed in equal proportions, and diluted 10-fold serially. The standard curves were generated using concentrations from 1 × 10^9^ to 1 × 10^2^ copies/µL. The results indicated that the slopes of the equations, correlation coefficients (R^2^), and amplification efficiencies (E) were −3.011, 0.998, and 114.862% for CCoV; −3.057, 0.999, and 112.377% for CRCoV; −2.902, 0.998, and 121.082% for CAV-2; and −3.143, 0.999, and 108.03% for CNV (Figure 1). An excellent relationship was shown between the initial concentration of template and the Ct value.

### 3.4. Specificity

The specificity of the quadruplex RT-qPCR was analyzed, and the results showed that only CCoV, CRCoV, CAV-2, and CNV had specific amplification profiles, while CDV (Snyder Hill, Onderstepoort, and CDV/R-20/8 strains), CPIV (NL-CPI-5, Cornell, and CPIV/A-20/8 strains), CPV (NL-35-D, 154, and CR86106 strains), CRV, and CanineCV had no amplification curve (Figure 2), indicating high specificity of the established assay.

### 3.5. Sensitivity

The concentrations of 400, 200, 100, and 50 copies/reaction were used to analyze the sensitivity of the established assay. The Ct values and hit rates of different concentrations of the plasmid constructs were assessed by Probit regression analysis (Table 3), and the LODs at 95% confidence interval (CI) were calculated. The LODs of p-CCoV, p-CRCoV, p-CAV-2, and p-CNV were determined to be 102.734 (93.835–116.766 at 95% CI), 104.777 (96.125–118.545 at 95% CI), 102.734 (93.835–116.766 at 95% CI), and 106.726 (98.194–120.638 at 95% CI) copies/reaction, i.e., 5.15, 5.25, 5.15, and 5.35 copies/μL, respectively (Figure 3), indicating high sensitivity in the established assay.

### 3.6. Repeatability

The repeatability was assessed using three concentrations of the mixture of p-CCoV, p-CRCoV, p-CAV-2, and p-CNV, i.e., 1 × 10^9^, 1 × 10^6^, and 1 × 10^3^ copies/µL. The results showed that the intra-assay and inter-assay CVs were 0.19–1.31% and 0.10–0.88%, respectively (Table 4), indicating the excellent repeatability of the established assay.

### 3.7. Test Results of Clinical Samples

The established quadruplex RT-qPCR was used to test the 1688 clinical samples collected during 2022–2024, and the positivity rates of CCoV, CRCoV, CAV-2, and CNV were 8.59% (145/1688), 8.65% (146/1688), 2.84% (48/1688), and 1.30% (22/1688), respectively (Table 5). The positivity rates of CCoV + CRCoV, CCoV + CAV-2, CCoV + CNV, CRCoV + CNV, CRCoV + CAV-2, CCoV + CAV-2 + CNV, CCoV + CRCoV + CAV-2, and CCoV + CRCoV + CNA co-infections were 2.37% (40/1688), 0.77% (13/1688), 0.06% (1/1688), 0.24% (4/1688), 0.24% (4/1688), 0.06% (1/1688), 0.53% (9/1688), and 0.06% (1/1688), respectively (Table 5).

The previously reported RT-qPCR methods [28,31,32,33] were also used to test the 1688 samples, and the positivity rates of CCoV, CRCoV, CAV-2, and CNV were 8.47% (143/1688), 8.53% (144/1688), 2.78% (47/1688), and 1.24% (21/1688), respectively (Table 5). The test results indicated that the clinical sensitivity and specificity of the established quadruplex RT-qPCR ranged from 95.45% to 100% and from 99.68% to 99.94% (Table 6). The overall coincidence rates between these assays were 99.53% (99.25–99.87% at 95% CI), 99.64% (99.23–99.84% at 95% CI), 99.94% (99.67–99.99% at 95% CI), and 99.88% (99.57–99.97% at 95% CI) for CCoV, CRCoV, CAV-2, and CNV, respectively (Table 7).

## 4. Discussion

Since the new emerging viruses and their continuous spread in canine population, canine viral diseases have become more complicated in recent years, with CIRDC and viral gastroenteritis posing serious threat to canine health. Dogs infected with CRCoV and CAV-2 often show clinical signs of respiratory syndromes [17,18,19,20], and CCoV and CNV cause gastroenteritis with similar symptoms [21,22,23,24], and it is hard to differentiate the diseases depending only on clinical signs, so differential detection of these pathogens is vital for accurate diagnosis and treatment of these diseases. Due to the advantages of qPCR/RT-qPCR, it is commonly used in veterinary laboratories for detection of viruses. To date, the previously reported qPCR/RT-qPCR [28,29,30,33] could detect the presence of only one or two of the following: CCoV, CRCoV, CAV-2, and CNV. In practice, CRCoV and CAV-2 cause respiratory signs, and CCoV and CNV cause gastrointestinal signs. At the same time, CRCoV can also cause enteritis, and CCoV can also cause respiratory signs [9,13]. Co-infections of these pathogens are common in clinical sick cases with respiratory and/or gastrointestinal signs [3,5,6,7], and it is necessary to detect and differentiate these pathogens for accurate diagnosis of these diseases. Effective diagnostic tools are important for the prevention, control, and treatment of CIRDC and viral gastroenteritis-associated diseases, so it is vital to establish a rapid and accurate assay to detect these four viruses. However, it is noteworthy that besides CRCoV and CAV-2, other important pathogens, such as CDV, canine herpesvirus (CHV), CPIV, canine influenza virus (CIV), and canine pneumovirus (CnPnV) can also induce CIRDC. Similarly, besides CCoV and CNV in intestinal infections, other important pathogens, such as CRV, CPV, canine astrovirus (CAstV), and canine kobuvirus (CaKoV), can cause similar symptoms of gastroenteritis. Therefore, it is important to conduct differential diagnoses of these diseases. In a previous report, a quadruplex RT-qPCR for the detection of CCoV, CRV, CPV, and CDV was successfully developed in our laboratory [34]. Since the co-infections of other multiple pathogens besides CCoV, CRV, CPV, and CDV in clinical sick cases, the quadruplex RT-qPCR for the detection of CCoV, CRCoV, CAV-2, and CNV was developed in this study to meet the needs of clinical testing and diagnosis. The positivity rates and co-infection rates of CCoV, CRCoV, CAV-2, and CNV in the 1688 clinical samples in this study (Table 4) confirmed the necessity of this quadruplex RT-qPCR assay.

The multiplex RT-qPCR has advantages of high specificity, sensitivity, and repeatability, and can rapidly, reliably, and accurately detect several pathogens at the same time in one reaction [25,26]. In this study, four pairs of specific primers and probes were designed for conserved regions of the CCoV M gene, CRCoV N gene, CAV-2 hexon gene, and CNV RdRp gene, and a quadruplex RT-qPCR was developed to simultaneously detect these four viruses in the same reaction. Other pathogens associated with CIRDC and gastroenteritis, including CDV, CPIV, CRV, CPV, and CanineCV, were used as differential controls for the specificity analysis of the established assay. Except for CCoV, CRCoV, CAV-2, and CNV with amplification curves, the absence of cross-reactivity with other canine viruses indicated that the developed assay had good specificity. The established assay showed high sensitivity, with LODs of about 10^2^ copies/reaction for CCoV, CRCoV, CAV-2, and CNV. The repeatability was excellent, with intra- and inter-assay CVs of 0.19–1.31% and 0.10–0.88%, respectively. Moreover, the 1688 clinical samples collected during 2022–2024 were tested using the developed and the reported reference assays [28,31,32,33]. The developed assay had high clinical sensitivity of 95.45–100% and clinical specificity of 99.68–99.94%, with agreement of 99.47% with the reference assays, confirming the clinical applicability of the established method.

The positivity rates of CCoV, CRCoV, CAV-2, and CNV in the 1688 clinical samples were 8.59%, 8.65%, 2.84%, and 1.30%, respectively. This indicated that these four viruses are widely prevalent in pet dogs in Guangxi province, China. Among them, the infections of CCoV and CRCoV were more serious than those of CAV-2 and CNV. Other reports indicated that infections with these four viruses were common [1,2,3,4]. Thieulent et al. [29] reported that 30.3% samples from Louisiana in the United States were positive for at least two respiratory viruses. Dong et al. [31] reported the positivity rates of CAV-2 and CRCoV in Jiangsu province of China were 1.08% and 2.97%, respectively. According to the review of Day et al. [3], CAV-2 have low infection rates, while CRCoV are commonly prevalent in European countries. Ma et al. [35] reported the 7.8% positivity rate of CNV in China. Especially, co-infections of these four viruses could not be ignored. In this study, the co-infection rates of CCoV + CRCoV, CCoV + CAV-2, CCoV + CNV, CRCoV + CNV, CRCoV + CAV-2, CCoV + CAV-2 + CNV, CCoV + CRCoV + CAV-2, and CCoV + CRCoV + CNA were 2.37%, 0.77%, 0.06%, 0.24%, 0.24%, 0.06%, 0.53%, and 0.06%, respectively. The co-infection of these viruses resulted in more severe pathological changes and clinical signs [5,6,7], which increases the damages of these pathogens.

Respiratory and gastrointestinal viruses can cause similar clinical signs and pathological changes, and co-infections of these viruses were common [5,6,7]. In recent years, the potential threat of animal diseases to human health has attracted increasing attention. As important pets in China, dogs are in close contact with humans, and the problem of cross-species transmission of human–canine viruses is becoming increasingly serious. CNV is an important pathogen of gastroenteritis and is the potential source of novel human NoVs through interspecies transmission [36]. According to the previous reports, dogs could also be carriers of human noroviruses [20,37]. Sick dogs show mild clinical signs of watery diarrhea and low mortality rate, which is similar to human norovirus infections. The detection of CNV in domestic sewage in Uruguay is a warning of the threat of human–dog cross-infection [38]. In addition, CCoV is prevalent in many countries around the world [39,40], and the zoonotic potentials of CCoV have attracted high attention [9,41,42,43,44]. Previous reports have indicated that humans can be infected by the new canine alphacoronavirus [41,42]. Some CCoV-like strains from humans might originate from recombinant CCoV variants, which obtained cross-transmission and became the zoonotic origin to humans [43,44]. Therefore, surveillance of these viruses with zoonotic potential is very urgent. A quadruplex RT-qPCR was developed in this study to rapidly and accurately detect these viruses and can be used to detect these viruses at an early stage of infection, which will help to prevent the spread of the virus and reduce the economic losses of these diseases.

## 5. Conclusions

In this study, a quadruplex RT-qPCR assay for the detection of CCoV, CRCoV, CAV-2, and CNV was successfully developed, showing high specificity, sensitivity, and repeatability. The assay could detect CCoV, CRCoV, CAV-2, and CNV simultaneously in one reaction, and could be used for the detection and monitoring of these canine diarrhea and respiratory viruses.

## Figures and Tables

**Figure 1 pathogens-13-01054-f001:**
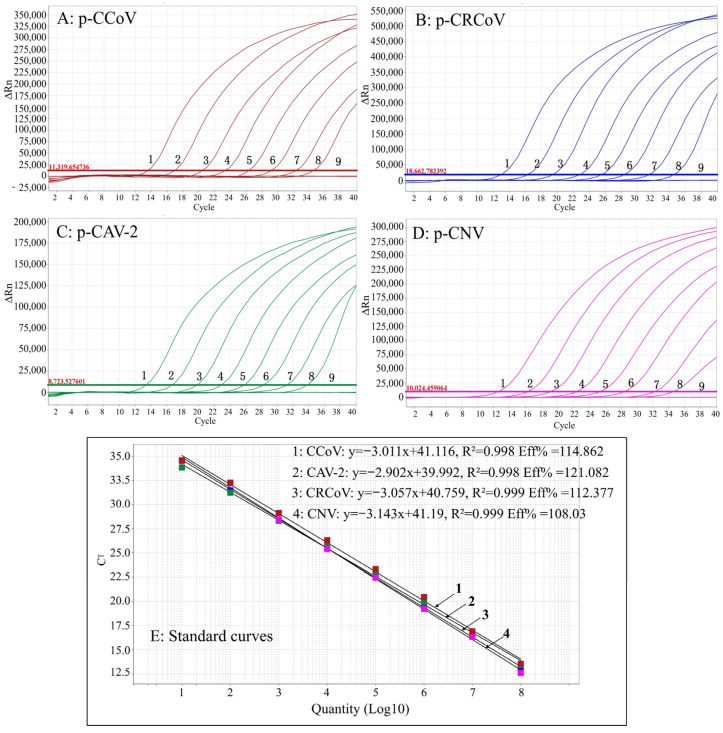
Amplification and standard curves of the quadruplex RT-qPCR. (**A**–**D**) Amplification curves of CCoV, CRCoV, CAV-2, and CNV; (**E**) standard curves. (1–8) The concentrations of plasmid constructs range from 1 × 10^9^ to 1 × 10^2^ copies/µL; (9) nuclease-free distilled water.

**Figure 2 pathogens-13-01054-f002:**
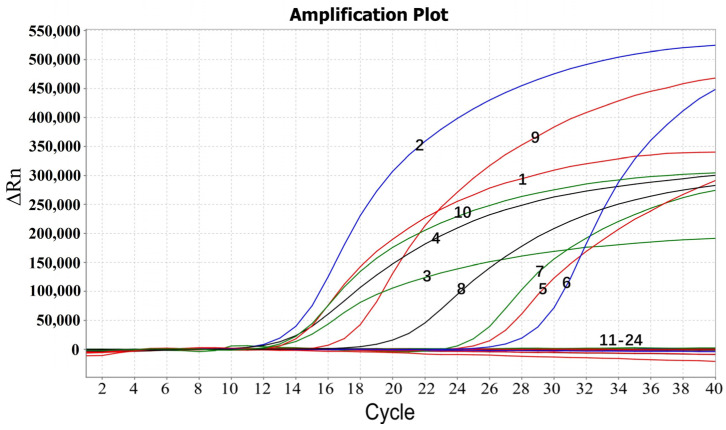
Specificity analysis of the quadruplex RT-qPCR. 1: p-CCoV; 2: p-CRCoV; 3: p-CAV-2; 4: p-CNV; 5: CCoV; 6: CRCoV; 7: CVA-2; 8: CNV; 9: CCoV (NL-18); 10: CAV-2 (Manhattan); 11–21: CDV (Snyder Hill), CPIV (NL-CPI-5), CPV (NL-35-D), CDV (Onderstepoort), CPV (154), CPIV (Cornell), CDV (CDV/R-20/8), CPIV (CPIV/A-20/8), CPV (CR86106), CRV, and CanineCV; 22–23: negative clinical samples; 24: distilled water.

**Figure 3 pathogens-13-01054-f003:**
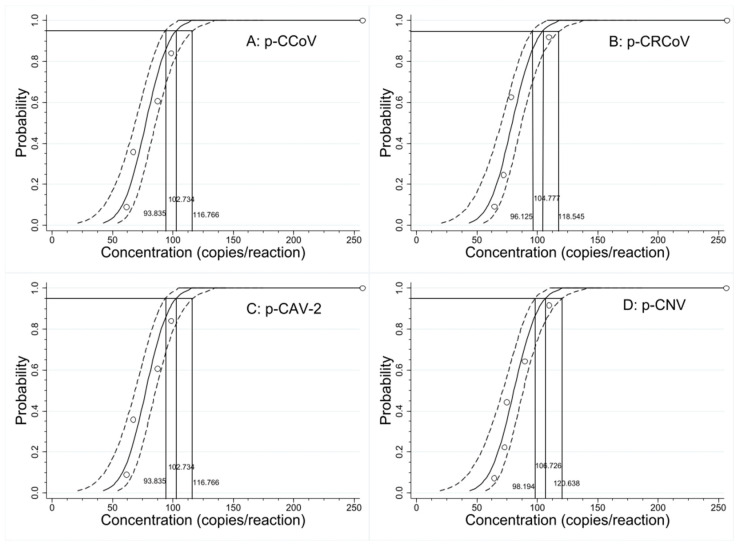
The sensitivity analysis by Probit regression analysis. The LODs of p-CCoV (**A**), p-CRCoV (**B**), p-CAV-2 (**C**), and p-CNV (**D**) were all about 1.0 × 10^2^ copies/reaction.

**Table 1 pathogens-13-01054-t001:** The designed primers and probes.

Name	Sequence (5′-3′)	Tm/°C	Genotype	Gene	Product/bp
CCoV (M)-F	GGTGGTATGAACATCGACAATT	57.3	CCoV-I CCoV-II	M	134
CCoV (M)-R	TTAGATTTTACATAGTAAGCCCATCC	56.0			
CCoV (M)-P	FAM-CGTAATGGTTGCATTACCTAGCAGGACCAT-BHQ1	65.6			
CRCoV (N)-F	TGGGTCGCTAGTAACCAGG	58.6	CRCoV	N	145
CRCoV (N)-R	TAACCCTGAGGGAGTACCG	56.5			
CRCoV (N)-P	ROX-CGATCGGGACCCAAGTAGCGATG-BHQ2	66.9			
CAV-2 (hexon)-F	CACAGAAATGCAGGACTCCG	57.6	CAV-2	hexon	119
CAV-2 (hexon)-R	TGAAAGACCACTCGTACGTG	58.5			
CAV-2 (hexon)-P	JOE-GCTTGGCAATGGCCGCTATTGCT-BHQ1	65.5			
CNV (RdRp)-F	CCAAGTTCGARGCCATGT	54.9	CNV-GVI CNV-GIV	RaRp	104
CNV (RdRp)-R	TTAGACGCCATCTTCATTCAC	54.8			
CNV (RdRp)-P	CY5- GCGAGATTGCGATCTCCCTCCCAC-BHQ3	65.8			

Note: For the primer and probe, F/R: forward/reverse primer; P: TaqMan probe. For primer sequences, degenerate base R = G/A.

**Table 2 pathogens-13-01054-t002:** The reaction system of the quadruplex RT-qPCR for CCoV, CRCoV, CAV-2, and CNV.

Ingredient	Volume/µL	Final Concentration/nΜ
CCoV (M)-F (20 pmol/µL)	0.3	300
CCoV (M)-R (20 pmol/µL)	0.3	300
CCoV (M)-P (20 pmol/µL)	0.3	300
CRCoV (N)-F (20 pmol/µL)	0.3	300
CRCoV (N)-R (20 pmol/µL)	0.3	300
CRCoV (N)-P (20 pmol/µL)	0.3	300
CAV-2 (hexon)-F (20 pmol/µL)	0.3	300
CAV-2 (hexon)-R (20 pmol/µL)	0.3	300
CAV-2 (hexon)-P (20 pmol/µL)	0.3	300
CNV (RdRp)-F (20 pmol/µL)	0.3	300
CNV (RdRp)-R (20 pmol/µL)	0.3	300
CNV (RdRp)-P (20 pmol/µL)	0.3	300
2× One-Step RT-PCR Buffer III	10	/
Ex Taq HS (5 U/µL)	0.4	/
PrimeScript RT Enzyme Mix II	0.4	/
RNA/DNA Template	2	/
Distilled Water	3.6	/
Total	20	/

**Table 3 pathogens-13-01054-t003:** The Ct values and hit rates.

Plasmid Construct	Concentration (Copies/Reaction)	Number of Samples	Quadruplex RT-qPCR	
			$\mathrm{Ct}\;(\bar{X}$)	Hit Rate (%)
p-CCoV	400	35	34.012	100
	200	35	34.591	100
	100	35	35.982	97.14
	50	35	ND	0
P-CRCoV	400	35	33.311	100
	200	35	34.198	100
	100	35	34.991	94.26
	50	35	ND	0
P-CAV-2	400	35	33.675	100
	200	35	34.836	100
	100	35	35.465	97.14
	50	35	ND	0
P-CNV	400	35	33.834	100
	200	35	34.903	100
	100	35	35.671	91.43
	50	35	ND	0

**Table 4 pathogens-13-01054-t004:** Repeatability of the quadruplex RT-qPCR.

Plasmid	Concentration (Copies/µL)	Concentration (Copies/Reaction)	Ct Values of Intra-Assay			Ct Values of Inter-Assay		
			$\bar{X}$	SD	CV%	$\bar{X}$	SD	CV%
p-CCoV	1 × 10^9^	2 × 10^10^	13.178	0.121	0.92	13.452	0.052	0.39
	1 × 10^6^	2 × 10^7^	23.283	0.045	0.19	23.264	0.096	0.41
	1 × 10^3^	2 × 10^4^	32.165	0.105	0.33	32.133	0.101	0.31
P-CRCoV	1 × 10^9^	2 × 10^10^	12.935	0.068	0.52	12.890	0.057	0.44
	1 × 10^6^	2 × 10^7^	22.506	0.296	1.31	22.721	0.123	0.54
	1 × 10^3^	2 × 10^4^	31.602	0.218	0.69	31.413	0.031	0.10
p-CAV-2	1 × 10^9^	2 × 10^10^	13.360	0.051	0.38	13.568	0.119	0.88
	1 × 10^6^	2 × 10^7^	22.827	0.153	0.67	22.880	0.074	0.32
	1 × 10^3^	2 × 10^4^	31.557	0.275	0.87	31.218	0.032	0.10
p-CNV	1 × 10^9^	2 × 10^10^	12.854	0.136	1.06	12.561	0.022	0.17
	1 × 10^6^	2 × 10^7^	22.476	0.238	1.06	22.367	0.053	0.24
	1 × 10^3^	2 × 10^4^	31.315	0.073	0.23	31.769	0.102	0.32

**Table 5 pathogens-13-01054-t005:** Test results of clinical samples using the developed and the reference assays.

Pathogen	Sample	Developed RT-qPCR		Reference RT-qPCR	
		Positive	Percentage (%)	Positive	Percentage (%)
Single Infection					
CCoV	1688	80	4.74%	79	4.68%
CRCoV	1688	92	5.45%	90	5.33%
CAV-2	1688	21	1.24%	24	1.42%
CNV	1688	15	0.89%	13	0.77%
Co-Infection					
CCoV + CRCoV	1688	40	2.37%	39	2.31%
CCoV + CAV-2	1688	13	0.77%	12	0.71%
CCoV + CNV	1688	1	0.06%	2	0.12%
CRCoV + CAV-2	1688	4	0.24%	2	0.12%
CRCoV + CNV	1688	4	0.24%	3	0.18%
CCoV + CAV-2 + CNV	1688	1	0.06%	1	0.06%
CCoV + CRCoV + CAV-2	1688	9	0.53%	8	0.47%
CCoV + CRCoV + CNV	1688	1	0.06%	2	0.12%
Total (Single + Co-Infection)					
CCoV	1688	145	8.59%	143	8.47%
CRCoV	1688	146	8.65%	144	8.53%
CAV-2	1688	48	2.84%	47	2.78%
CNV	1688	22	1.30%	21	1.24%

**Table 6 pathogens-13-01054-t006:** The clinical sensitivity and specificity of the quadruplex RT-qPCR.

Established RT-qPCR		Reference RT-qPCR		Total	Clinical Sensitivity (95% CI)	Clinical Specificity (95% CI)
		Positive	Negative			
CCoV	Positive	140	5	145	97.90% (94.01–99.28%)	99.68% (99.24–99.86%)
	Negative	3	1540	1543		
	Total	143	1545	1688		
CRCoV	Positive	142	4	146	99.61% (95.08–99.62%)	99.74% (99.34–99.90%)
	Negative	2	1540	1542		
	Total	144	1544	1688		
CAV-2	Positive	47	1	48	100% (92.44–100%)	99.94% (99.66–99.99%)
	Negative	0	1640	1640		
	Total	47	1641	1688		
CNV	Positive	21	1	22	95.45% (78.20–99.19%)	99.94% (99.66–99.99%)
	Negative	1	1665	1666		
	Total	22	1666	1688		

**Table 7 pathogens-13-01054-t007:** Agreements between the developed and the reference RT-qPCR.

Method	Positive Sample			
	CCoV	CRCoV	CAV-2	CNV
Established RT-qPCR	145/1688 (8.59%)	146/1688 (8.65%)	48/1688 (2.84%)	22/1688 (1.30%)
Reference RT-qPCR	143/1688 (8.47%)	144/1688 (8.53%)	47/1688 (2.78%)	21/1688 (1.24%)
Agreements (95% CI)	99.53% (92.25–99.87%)	99.64% (99.23–99.84%)	99.94% (99.67–99.99%)	99.88% (99.57–99.97%)

## Data Availability

The raw data can be obtained from the corresponding authors upon appropriate request.

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
