# Peer review of "Establishment of a Quadruplex RT-qPCR for the Detection of Canine Coronavirus, Canine Respiratory Coronavirus, Canine Adenovirus Type 2, and Canine Norovirus"

_pathogens, 2024, doi:10.3390/pathogens13121054_

Round 1

Reviewer 1 Report

Comments and Suggestions for Authors

In this study, Shi et al. developed a quadruplex reverse transcription quantitative polymerase chain reaction (RT-qPCR) assay for the detection of Canine Coronavirus (CCoV), Canine Respiratory Coronavirus (CRCoV), Canine Adenovirus type 2 (CAV-2), and Canine Norovirus (CNV). The specificity, sensitivity, and repeatability of the established assay were evaluated. Subsequently, the assay was applied to test 1,688 clinical samples collected from veterinary hospitals in Guangxi Province, China, between 2022 and 2024 to validate its clinical applicability. In conclusion, a quadruplex RT-qPCR assay with high sensitivity, specificity, and repeatability was developed for the rapid and accurate detection of CCoV, CRCoV, CAV-2, and CNV. While the design of this study is scientifically sound, several issues must be addressed.

 1. The introduction should emphasize the necessity of establishing RT-qPCR for the detection of CCoV, CRCoV, CAV-2, and CNV.

2. The reference strain requires the provision of a genome accession number.

3. The detailed information of clinical samples needs to be listed.

4. The target genes of the primers need to be reflected in Table 1.

5. Section 3.1 lacks results for standard plasmid constructs and their identification; please provide this information.

Author Response

The Cover Letter

November 21, 2024

Dear editor,

The manuscript has been revised carefully according to the reviewers’ suggestions. The details were as follows.

Reviewer #1

Comments and Suggestions for Authors

In this study, Shi et al. developed a quadruplex reverse transcription quantitative polymerase chain reaction (RT-qPCR) assay for the detection of Canine Coronavirus (CCoV), Canine Respiratory Coronavirus (CRCoV), Canine Adenovirus type 2 (CAV-2), and Canine Norovirus (CNV). The specificity, sensitivity, and repeatability of the established assay were evaluated. Subsequently, the assay was applied to test 1,688 clinical samples collected from veterinary hospitals in Guangxi Province, China, between 2022 and 2024 to validate its clinical applicability. In conclusion, a quadruplex RT-qPCR assay with high sensitivity, specificity, and repeatability was developed for the rapid and accurate detection of CCoV, CRCoV, CAV-2, and CNV. While the design of this study is scientifically sound, several issues must be addressed.

  1. The introduction should emphasize the necessity of establishing RT-qPCR for the detection of CCoV, CRCoV, CAV-2, and CNV.

Response: We agree to the reviewer’s suggestion. The necessity of establishing RT-qPCR for the detection of CCoV, CRCoV, CAV-2, and CNV has been emphasized in the revised manuscript. Please see Lines 40-45 and 73-79 in the revised manuscript.

  1. The reference strain requires the provision of a genome accession number.

Response: We agree to the reviewer’s suggestion. Only some of the reference strains can be obtained their genome accession numbers from GenBank. Please see Lines 95-97 in the revised manuscript.

  1. The detailed information of clinical samples needs to be listed.

Response: We agree to the reviewer’s suggestion. The detailed information of clinical samples has been added in the revised manuscript. Please see Lines 106-113 in the revised manuscript.

  1. The target genes of the primers need to be reflected in Table 1.

Response: We agree to the reviewer’s suggestion. The target genes of the primers have been added to Table 1. Please see Table 1 in the revised manuscript.

  1. Section 3.1 lacks results for standard plasmid constructs and their identification; please provide this information.

Response: We agree to the reviewer’s suggestion. The information of the results for standard plasmid constructs and their identification has been added in the revised manuscript. Please see Lines 210-219 in the revised manuscript.

Best regards,

Kaichuang Shi

Reviewer 2 Report

Comments and Suggestions for Authors

          The authors developed a quadruplex RT-qPCR aasay for the detection of four canine pathogens: Canine Coronavirus, Canine Respiratory Coronavirus, Canine 3 Adenovirus Type 2, and Canine Norovirus. The study is rigorous. Some concerns should be addressed.

1.    The authors describe a limit of detection of 100 copies/reaction. It would be important to infer from this the limit of detection in terms of copies/ml.

2.    Lines 116-120. Was the totality of the genome sequences available at GenBank downloaded for this study? If not, what was the criteria?

3.    Supplementary material Figure S1: There are some mismatch (only one nucleotide) for two of the four probes, for one isolate each time, whose sequence is available on GenBank. Do the authors expect that the probe will detect the target sequence in these cases? Were all the genotypes for each virus aligned in Figure S1?

4.    Discussion line 281. what is the interest in detecting in a same reaction both a respiratory and a gastrointestinal disease in dogs?

5.    The authors published recently a similar study of quadruplex detection of viral pathogens in dogs, with canine coronavirus but detecting other viruses: Shi et al. A Quadruplex Reverse Transcription Quantitative Polymerase Chain Reaction for Detecting Canine Coronavirus, Canine Rotavirus, Canine Parvovirus, and Canine Distemper Virus. Microbiol. Res. 202415(2), 746-761. https://doi.org/10.3390/ microbiolres15020049. What is the rationale of these combinations of viral infection being detected in quadruplex real time format?

6.    From points 4 and 5, it would be advisable to support (with more epidemiological data for example) the rationale of the new combination proposed in this study.

Author Response

The Cover Letter

November 21, 2024

Dear editor,

The manuscript has been revised carefully according to the reviewers’ suggestions. The details were as follows.

Comments and Suggestions for Authors

          The authors developed a quadruplex RT-qPCR assay for the detection of four canine pathogens: Canine Coronavirus, Canine Respiratory Coronavirus, Canine Adenovirus Type 2, and Canine Norovirus. The study is rigorous. Some concerns should be addressed.

  1. The authors describe a limit of detection of 100 copies/reaction. It would be important to infer from this the limit of detection in terms of copies/ml.

Response: We agree to the reviewer’s suggestion. The limit of detection has also been described in copies/μL beside copies/reaction in the revised manuscript. Please see Lines 260-264 in the revised manuscript.

  1. Lines 116-120. Was the totality of the genome sequences available at GenBank downloaded for this study? If not, what was the criteria?

Response: We agree to the reviewer’s suggestion. Only part of the genome sequences available at GenBank were downloaded. These representative strains of CCoV, CRCoV, CAV-2, and CNV were selected basing on their origination from different countries, and isolation of different dates. This content has been added in the revised manuscript. Please see Lines 134-136 in the revised manuscript.

  1. Supplementary material Figure S1: There are some mismatch (only one nucleotide) for two of the four probes, for one isolate each time, whose sequence is available on GenBank. Do the authors expect that the probe will detect the target sequence in these cases? Were all the genotypes for each virus aligned in Figure S1?

Response: We agree to the reviewer’s suggestion. In this study, the primers and probes were designed for universal detection of different genotypes of CCoV, CRCoV, CAV-2, and CNV. In Figure S1, all the genotypes of CCoV, CRCoV, and CAV-2 were aligned, and genotypes GVI and GIV of CNV were aligned. The primers and probes were designed to detect all the genotypes of CCoV, CRCoV, and CAV-2, and genotypes GVI and GIV of CNV in this study. Please see Figure S1 in the revised manuscript.

  1. Discussion line 281. what is the interest in detecting in a same reaction both a respiratory and a gastrointestinal disease in dogs?

Response: We agree to the reviewer’s suggestion. The reasons of detecting in a same reaction both a respiratory and a gastrointestinal disease in dogs have been discussed in the revised manuscript. Please see Lines 304-309 in the revised manuscript.

  1. The authors published recently a similar study of quadruplex detection of viral pathogens in dogs, with canine coronavirus but detecting other viruses: Shi et al. A Quadruplex Reverse Transcription Quantitative Polymerase Chain Reaction for Detecting Canine Coronavirus, Canine Rotavirus, Canine Parvovirus, and Canine Distemper Virus. Microbiol. Res. 2024, 15(2), 746-761. https://doi.org/10.3390/ microbiolres15020049. What is the rationale of these combinations of viral infection being detected in quadruplex real time format?

Response: We agree to the reviewer’s suggestion. In a previous report, a quadruplex RT-qPCR for the detection of CCoV, CRV, CPV, and CDV was successfully developed in our laboratory. Since the co-infections of other multiple pathogens besides CCoV, CRV, CPV, and CDV in clinical sick cases, the quadruplex RT-qPCR for the detection of CCoV, CRCoV, CAV-2, and CNV was developed in this study to meet the needs of clinical testing and diagnosis. Please see Lines 317-324 in the revised manuscript.

  1. From points 4 and 5, it would be advisable to support (with more epidemiological data for example) the rationale of the new combination proposed in this study.

Response: We agree to the reviewer’s suggestion. Since co-infections of multiple pathogens are common in clinical sick cases with respiratory and/or gastrointestinal signs, it is necessary to detect and differentiate these pathogens for accurate diagnosis of these diseases. Please see Lines 304-309 and 317-324 in the revised manuscript.

Best regards,

Kaichuang Shi

Reviewer 3 Report

Comments and Suggestions for Authors

In this study, the authors developed a multiplex system for detecting 4 pathogens simultaneously. The manuscript is well written. 

The preparation of clinical samples and DNA extraction should be described in detail. In addition, to exclude inefficient extraction of nucleic acids or the potential presence of inhibitors in the extracted nucleic acids that could affect PCR amplification, the reviewer also suggests adding an internal control cGAPDH and developing or using the described PCR detection system (for example as in [31]).

Typos:

Please add a legend to Table 1: F/R: forward/reverse primer; P: TaqMan probe.

Some data is repeated. For example, the initial concentration of plasmids in the paragraph 2.6 (Materials and Methods) and in the results (3.1. Construction of Standard Plasmid Constructs). it is preferable to leave this data only in one of the paragraphs.

Line 144: The positive clones were cultured FOR 22-24 h at 37 °C

Line 214: Did the authors mean Figure 1?

Line 226: Figure 3 should be Figure 2 and so on (Figure 4 should be Figure 3.

Line 235: «were used to analyze»

Author Response

The Cover Letter

November 21, 2024

Dear editor,

The manuscript has been revised carefully according to the reviewers’ suggestions. The details were as follows.

Comments and Suggestions for Authors

In this study, the authors developed a multiplex system for detecting 4 pathogens simultaneously. The manuscript is well written.

  1. The preparation of clinical samples and DNA extraction should be described in detail.

Response: We agree to the reviewer’s suggestion. The preparation of clinical samples and DNA extraction have been described in detail in the revised manuscript. Please see Lines 105-113 and 114-128.

  1. In addition, to exclude inefficient extraction of nucleic acids or the potential presence of inhibitors in the extracted nucleic acids that could affect PCR amplification, the reviewer also suggests adding an internal control cGAPDH and developing or using the described PCR detection system (for example as in [31]).

Response: We agree to the reviewer’s suggestion. In this study, the internal control cGAPDH and the described PCR detection system were not used. The recombinant plasmid constructs, the positive clinical samples, the negative clinical samples, and nuclease-free distill water were used as positive control, and negative control, respectively. Please see Lines 183-189 in the revised manuscript.

Typos:

  1. Please add a legend to Table 1: F/R: forward/reverse primer; P: TaqMan probe.

Response: We agree to the reviewer’s suggestion. This content has been added in Table 1 in the revised manuscript. Please see Lines 146-147 in the revised manuscript.

  1. Some data is repeated. For example, the initial concentration of plasmids in the paragraph 2.6 (Materials and Methods) and in the results (3.1. Construction of Standard Plasmid Constructs). it is preferable to leave this data only in one of the paragraphs.

Response: We agree to the reviewer’s suggestion. The data in the paragraph 2.6 has been deleted. Please see Lines 165-167 and 219-221.

  1. Line 144: The positive clones were cultured FOR 22-24 h at 37 °C

Response: We agree to the reviewer’s suggestion. “for” has been added. Please see Line 161 in the revised manuscript.

  1. Line 214: Did the authors mean Figure 1?

Response: We agree to the reviewer’s suggestion. It is Figure 1. Please see Line 236 in the revised manuscript.

  1. Line 226: Figure 3 should be Figure 2 and so on (Figure 4 should be Figure 3).

Response: We agree to the reviewer’s suggestion. It is Figure 2 and 3. Please see Lines 248 and 263 in the revised manuscript.

  1. Line 235: «were used to analyze»

Response: We agree to the reviewer’s suggestion. Change “analyzed” to “analyze”. Please see Line 257 in the revised manuscript.

Best regards,

Kaichuang Shi

Round 2

Reviewer 2 Report

Comments and Suggestions for Authors

  The authors addressed the concerns.

Reviewer 3 Report

Comments and Suggestions for Authors

The authors answered the questions of the rewiever